# NASA GeneLab Platform Utilized for Biological Response to Space Radiation in Animal Models

**DOI:** 10.3390/cancers12020381

**Published:** 2020-02-07

**Authors:** J. Tyson McDonald, Robert Stainforth, Jack Miller, Thomas Cahill, Willian A. da Silveira, Komal S. Rathi, Gary Hardiman, Deanne Taylor, Sylvain V. Costes, Vinita Chauhan, Robert Meller, Afshin Beheshti

**Affiliations:** 1RadBioX Services LLC, Okemos, MI 48864, USA; RadBioX@RadBioX.com; 2Consumer and Clinical Radiation Protection Bureau, Health Canada, Ottawa, ON K1A-1C1, Canada; robert.stainforth@cananda.ca (R.S.); vinita.chauhan@canada.ca (V.C.); 3KBR, NASA Ames Research Center, Moffett Field, CA 94035, USA; j_miller@lbl.gov; 4School of Biological Sciences & Institute for Global Food Security, Queens University Belfast, Belfast BT9 5DL, UK; tcahill01@qub.ac.uk (T.C.); W.daSilveira@qub.ac.uk (W.A.d.S.); 5Department of Biomedical Informatics, The Children’s Hospital of Philadelphia, Philadelphia, PA 19104, USA; rathik@email.chop.edu; 6Department of Medicine, Medical University of South Carolina, Charleston, SC 29425, USA; G.Hardiman@qub.ac.uk; 7The Center for Mitochondrial and Epigenomic Medicine, The Children’s Hospital of Philadelphia, Philadelphia, PA 19104, USA; taylordm@email.chop.edu; 8The Department of Pediatrics, Perelman School of Medicine, University of Pennsylvania, Philadelphia, PA 19104, USA; 9NASA Ames Research Center, Space Biosciences Division, Moffett Field, CA 94035, USA; sylvain.v.costes@nasa.gov; 10Department of Neurobiology and Pharmacology, Morehouse School of Medicine, Atlanta, GA 30310, USA; rmeller@msm.edu

**Keywords:** GeneLab, NASA, space radiation, HZE, galactic cosmic rays, transcriptomics, dosimetry, radiation

## Abstract

**Background:** Ionizing radiation from galactic cosmic rays (GCR) is one of the major risk factors that will impact the health of astronauts on extended missions outside the protective effects of the Earth’s magnetic field. The NASA GeneLab project has detailed information on radiation exposure using animal models with curated dosimetry information for spaceflight experiments. **Methods:** We analyzed multiple GeneLab omics datasets associated with both ground-based and spaceflight radiation studies that included in vivo and in vitro approaches. A range of ions from protons to iron particles with doses from 0.1 to 1.0 Gy for ground studies, as well as samples flown in low Earth orbit (LEO) with total doses of 1.0 mGy to 30 mGy, were utilized. **Results:** From this analysis, we were able to identify distinct biological signatures associating specific ions with specific biological responses due to radiation exposure in space. For example, we discovered changes in mitochondrial function, ribosomal assembly, and immune pathways as a function of dose. **Conclusions:** We provided a summary of how the GeneLab’s rich database of omics experiments with animal models can be used to generate novel hypotheses to better understand human health risks from GCR exposures.

## 1. Introduction

Space is a hostile environment, and human space travel will entail effects at many levels of biological organization leading to related physiological effects. One of the major environmental factors in space is radiation. Ionizing radiation in space is composed of a complex mixture of particles including protons from the sun and galactic cosmic rays (GCRs) originating from outside the solar system. The GCR is composed of atomic nuclei, including a component of heavy charged and energetic (HZE) particles that are relatively few in number but highly ionizing. A major risk-factor for human spaceflight is exposure GCR HZE that can cause DNA damage that is unrepaired or mis-repaired [1]. 

Current research has shown measurable impact on in several areas affecting human health bone loss [2], skeletal muscle atrophy [3,4,5], reproductive hazards in women and men [6], cognitive/behavioral changes [7], central nervous system (CNS) decrements [8,9], and changes in cardiovascular physiology [10]. The possibility that space radiation exposure to increase cardiovascular disease risk requires additional validation similar to the ongoing observations of increased cardiac impairment observed in radiotherapy-treated cancer survivors [11,12]. Spaceflight has also been shown to have an impact on established cancer hallmarks [13], possibly increasing carcinogenic risk [14]. 

At the molecular level, space radiation from high energy photons and atomic nuclei can cause damage to cellular components, in particular deoxyribonucleic acid (DNA). Damaged DNA, which is inadequately repaired, can led to mutations and chromosomal aberrations that have potential to propagate into RNA, potentially leading to cancer. Simulations using ground-based datasets representative of a combined space radiation and microgravity exposure have demonstrated that oxidative stress processes and mitochondrial dysfunctions are important contributors to biological effects from spaceflight. Simulated microgravity studies have demonstrated an inhibitory effect on cell cycle and cell proliferation [15]. These studies show the value of research centered on understanding the totality of space exposures [16]. A multitude of other biological processes that are affected include increased genomic instability and mutation rates [17], immune dysregulation [18], and deregulated cellular energetics [19]. Conflicting results on cellular stemness [20] have been reported with effects such as elongation of telomere and altered telomerase function that require clarification [21,22]. 

A substantial amount of work has also been generated in the area of omics. For several decades, omics animal-based studies have been performed to understand the human health risks due to spaceflight missions using the International Space Station, the space shuttle, biosatellites, and ground-based experiments. NASA’s growing GeneLab omics database now hosts almost 250 experimental studies on spaceflight exposure and analogous ground-based experiments from microbes, plants, animal models, and human/animal-derived cell cultures [23]. These data are available as an open-access resource for single- and multi-omic analyses on microarray, RNA-sequencing, bisulfite-sequencing, proteomic, metabolomic, and metagenomic datasets. Although these data are targeted to better understand the risks associated with human spaceflight, these experiments also demonstrate a unique stress that provides a better understanding of adverse human health effects due to alterations in cellular behavior owing to factors such as microgravity, altered carbon dioxide levels, psychological stressors, and damaging ionizing radiation exposures from GCR and protons in the Earth’s radiation belts. These datasets, and in particular the in vivo animal models used, present a unique experimental setting to study the cellular and host-level interactions that impact human health.

In this study, a meta-analysis of these accumulated transcriptomic datasets in GeneLab relevant to spaceflight was undertaken to uncover potential dose-dependent biological responses that will help in understanding the human health risks experienced in the spaceflight environment. The studies identified were from human- and animal-derived tissues or cells exposed to protons and GCR HZEs at doses ranging from 0.1 to 1.0 Gy for ground studies and using samples flown to the International Space Station or shuttle missions with total doses between 1.0 and 30 mGy. Ground experiments are a well-established analog for space flight, but are limited in that they are relatively high dose, high dose rate, and in most cases use single particle species, in contrast to the low dose, low dose rate mixed radiation in space. Further work is needed to elucidate platform-dependent differences (mixed fields now being offered at the NASA Space Radiation Laboratory [24] are a step in this direction). The datasets were pooled and subjected to gene set enrichment analysis (GSEA) to identify clusters of gene sets and associated biological processes. The results from this analysis highlight novel cellular responses that may be relevant contributors to the combined effects of microgravity and space radiation and therefore critical drivers of health risks from space radiation. 

## 2. Results

Using primarily in vivo animal models and their associated curated datasets available from the NASA GeneLab’s omics database, an examination of biological responses as a function of radiation dose was made using transcriptional and sequencing data. A total of 30 experimental conditions from 25 GeneLab Data System (GLDS) datasets were collected (Table 1). Multiple datasets were included from human or murine cell cultures or animal tissues from several anatomical sites (adrenal gland, blood, bone marrow, brain, breast, heart, cartilage, skin, eye, hair follicle, kidney, liver, lung, muscle, spleen, or thymus). For the simulated space radiation ground studies (GLDS-73, -80, -109, and -117) information related to energy, linear energy transfer (LET, a measure of energy deposition at the cellular and molecular scale), dose rate, particle charge, and particle mass are available and can be viewed in Appendix A. The dose data for all spaceflight experiments, with the exception of GLDS-111, were from instruments available on the Space Shuttle and the International Space Station. Analysis was completed by subject matter experts from the Space Radiation Analysis Group (SRAG), Johnson Space Center, NASA. The provenance of data from GLDS-111 (Bion-M1) is detailed in the environmental data section of the GLDS for Bion-M1. The doses and dose rates for ground-based exposures are from the NASA Space Radiation Laboratory with uncertainties that are less than 2% [24].

### 2.1. Changes in RNA-Specific Error Rates Comparing Spaceflight Versus Ground Control Samples

DNA mutations caused by ionizing radiation have been shown to propagate onto RNA during translation, unless they are adequately removed by transcription-coupled repair. Higher doses of radiation exposure to a cell will result in greater accumulated damage, up to a point where the cellular repair mechanisms are overwhelmed. However, lower levels of damage may be replicated into RNA with aberrant effects on translation and cellular function. Thus, RNA-sequencing data using mice liver samples from two experiments were analyzed to identify potential error rates due to spaceflight. Analysis of the GLDS-168 dataset showed no difference in overall mapping error rates across the entire genome (approximately 0.5%). Further investigation of the mismatches with respect to each chromosome identified error rates to be near 0.5%; however, there was no difference between the spaceflight and ground control groups (Figure 1A,B). Interestingly, there were differential error rates across some chromosomes, in particular the sex-associated chromosomes, and error rates for genes associated with mitochondrial DNA were lower than genomic DNA, supporting dynamic DNA damage repair in mitochondria (reviewed in [53]). Further analysis revealed increased rates of substitution in specific base sequences in GLDS-168 samples (Figure 1C) but not in the GLDS-242 dataset (Figure 1D). Specific base substitutions showed an increase in A>G and C>T substitutions in only GLDS-168 samples. We also observed a significant increase in CGT and GTA errors but a decrease in CAC and GTG errors in spaceflight animals versus ground controls (all corrected *p* < 0.05; Figure 1E). In contrast the GLDS-242 dataset showed an overall reduction, though non-significant, in specific base errors with a significant change in substitutions (*p* < 0.05, corrected) in spaceflight animals compared to ground controls (Figure 1D,F).

### 2.2. Comparison of Multiple Genelab Datasets Using Principal Component Analysis 

Each individual transcriptome dataset was analyzed by comparing flight/radiation exposure versus ground/control treatment. Utilizing standard bioinformatics pipelines for RNA-sequencing and microarray analyses, we obtained expression data for each dataset for use in gene set enrichment analysis (GSEA) [54]. The results indicated a wide variance of significantly regulated gene sets per dataset (Figure 2). When considering a false discovery rate (FDR) < 0.05, we observed some datasets with a low number of significantly regulated gene sets, whereas others had a high amount of significant results for the three GSEA molecular signature database collections (i.e., C5, C2, and C6). Thus, for further analysis, individual gene sets were considered for analysis if they had a FDR < 0.25 in 50% or more of the total datasets examined. Then, using Cytoscape’s enrichment mapper, these gene sets were grouped into networks and further examined using the auto-annotate feature to cluster similar gene sets on the basis of biological similarity, referred to as “annotated gene sets”. 

To observe if any trends or batch issues arise from this type of analysis, we compared the normalized enrichment scores (NES) for all the gene sets resulting from each molecular signature database collection (Figure 3 and Appendix A). Principal component analysis (PCA) plots indicated that there were no batch issues occurring globally for either the pre- or post-annotated gene sets as a function of dose (Figure 3). In addition, we did not observe any tissue dependence or clustering for the pre- or post-annotated gene sets (Appendix A). Lastly, these samples were also clustered with t-distributed stochastic neighbor embedding (t-SNE) plots to observe whether different methods would produce conflicting results (Appendix A). The t-SNE plots also provided additional validation that no batch issues are observed with this type of analysis when considering dose or tissue dependence.

To compare the number of genes sets with the annotated gene sets, we provided heatmaps and clustering for all the datasets based on the NES for each molecular signature database collection for the analysis with the FDR < 0.25 (Figure 4). As expected, the annotated gene sets provided fewer gene sets, enabling a more streamlined analysis downstream. The overall pattern of the datasets between the annotated gene sets and all the genes sets did not change. This indicates that any analysis on the annotated gene sets should provide similar results for all the gene sets.

### 2.3. Transcriptome Evaluation of Biological Functions Impacted by Spaceflight Exposure in Animal Models 

To observe the impact of radiation exposure, transcriptional data from experiments spanning 1 to 1000 mGy doses in animal models and cell cultures were examined for alteration in multiple biological pathways. This analysis combined statistically significant biological pathways that resulted from GSEA analysis for each individual dataset [54]. This included the following three major molecular signatures database collections: (1) the Gene Ontology (GO; C5) gene sets from biological processes, cellular components, and molecular functions; (2) the curated dataset (C2) gene sets from chemical and genetic perturbations and canonical pathways such as the BioCarta, Kyoto Encyclopedia of Genes and Genomes (KEGG), Pathway Interaction Database (PID), and Reactome pathway databases; and (3) the oncogenic signatures (C6) gene sets representing major pathways altered in cancer.

#### 2.3.1. GSEA C5: Gene Ontology Results

Using Cytoscape’s enrichment mapper, 63 statistically significant GSEA gene sets from the Gene Ontology (C5) database had a FDR < 25% in 50% or more of the GeneLab datasets analyzed. Using the auto-annotate clustering tool, these 63 gene sets represented 7 major biological pathways including: (1) ribosome assembly, (2) mitochondrial function, (3) interferon-gamma response, (4) chemotaxis, (5) cellular differentiation, (6) extracellular organization, and (7) cytokine production (Figure 5A). Each of these clusters was driven by an overrepresented enrichment of genes related to these pathways.

In addition to understanding common biological pathways disrupted by spaceflight and ground-based radiation exposure, these results were also used to examine possible dose-dependent trends. Using a *k*-means clustering of the heatmaps representing each clustering nominal enrichment scores, six major annotated clusters were overwhelmingly represented this data, including (1) mitochondrial pathways, (2) leukocyte and adaptive immune response, (3) wound healing and VEGF pathways, (4) ribosome, (5) lipid pathways, and (6) mitotic nuclear division (Figure 5B). The nominal enrichment score (NES) for these annotated gene sets was plotted from 1mGy to 1000 mGy and fitted using a generalized additive model (GAM) smoothing plot [55,56] (Figure 5C). Indeed, there was an observed activation as a function of dose for the mitochondrial pathway, whereas the clusters for the ribosome, lipid pathways, and mitotic nuclear division were generally inactivated. A more uncertain trend was observed for the immune related clusters involving leukocyte and adaptive immune response as well as wound healing and VEGF pathways, demonstrating the complex nature of normal tissue responses to inflammation and immune responses. 

We also performed an analysis with less stringent statistics, involving a larger set of gene sets, to observe whether other functions or pathways would appear. We performed the same analysis as above but included significant GSEA gene sets from the Gene Ontology (C5) database that had an FDR < 0.05 in only three of the GeneLab datasets analyzed. Although the FDR was set at a lower threshold, the lower number of required datasets with an FDR < 0.05 threshold (Figure 2) would result in more gene sets for the downstream analysis. We saw this as true when observing the number of gene sets (Appendix A). Some *k*-means clusters were too dense and were split into sub-clusters (Appendix A). The GAM fits were provided for each cluster and sub-cluster (Appendix A). As expected, this less stringent analysis approach produced less statistically significant fits with more noise confounding the data. Interestingly, some new trends appeared that can be related to space radiation-induced health risks, including decreases in muscle-related pathways as a function of increasing dose (Appendix A) and decreases in chromosome pathways with increasing dose (Appendix A). This inclusive analysis strategy demonstrated that analyzing the data in many different ways can reveal relevant results when that can easily be missed with only one methodology and overly strict statistical cutoffs. 

#### 2.3.2. GSEA C2: Curated Dataset Results

To expand these results, the canonical pathways (C2) molecular signatures database collection was examined in a similar manner to the gene ontology collection (C5) above. For this analysis, there were 30 total datasets as opposed to 28 for the C5 gene set analysis. We determined 411 significantly regulated C2 gene sets using Cytoscape’s enrichment mapper with an FDR ≤ 25% in 50% of the GeneLab datasets analyzed. Due to the C2 GSEA database containing more gene sets, a larger number of gene sets and auto-annotated terms was expected. From these terms, we observed several key pathways, including mitotic and cell cycle pathways, cancer-related pathways, hypoxia-related pathways, interferon pathways, and pathways related to radiation impact on cells (Figure 6A). Cell cycle pathways [57], hypoxia [58], interferon dysregulation [59], and cancer [60,61,62] have all been observed in previous research studies as dysregulated due to spaceflight. 

We performed a similar analysis with C2 genes sets as we did with the C5 dose-dependent analysis (Figure 5 and Figure 6). Using *k*-means clustering for heatmaps representing each clustering nominal enrichment scores, 15 major annotated clusters overwhelmingly represented these data, including (1) hypoxia, MYC, and phospholipid pathways; (2) EGF, TNF, and hematopoietic pathways; (3) E2F3, EGFR2, p53, and cell cycle pathways; (4) luminal, miR-21 targets, KRAS, and transformation pathways; (5) RB1, metastasis, mitotic division, and progenitor pathways; (6) hematopoiesis, interferons, and methylation pathways; (7) obesity-, oxidation-, and adipogenic-related pathways; (8) mitochondrial and adipogenesis pathways; (9) complex EIF binding; (10) progesterone; (11) apoptosis and phospholipid pathways; (12) inflammatory, invasion, NOTCH, and HNF1a pathways; (13) carcinoma, hypoxia, and IL1a pathways; (14) PKD1, ZMPSTE24, and cardiac dysfunctions; and (15) ICP with H3K27ME3 pathway (Figure 6B,C). Similar to the C5 GO analysis, we observed an increase in mitochondrial pathways as a function of dose. Observing this trend with another pathway database provided additional confidence with these results. A clear decrease with cardiac dysfunctions, PKD1, and ZMPSTE24 was revealed as a function of dose. Interestingly, deficiencies or decreases in both *PKD1* [63] and *ZMPSTE24* [64] expression have been previously shown to cause cardiac dysfunction. These observed biological functions seem to follow a trend that may have be missed by manual analysis.

Lastly, less stringent statistics with an FDR < 0.05 in only three of the GeneLab Datasets was also analyzed for C2 gene sets (Appendix A). Like the C5 analysis, more gene sets were clustered per group after *k*-means clustering (Appendix A), leading to three clusters requiring additional *k*-means sub-clustering analysis (Appendix A). This less statistically stringent analysis on C2 produced additional trends through the GAM fits that were not observed in the more conservative analysis (Appendix A). A clear decrease in TNF pathways was observed with increasing dose (Appendix A), and a novel pathway related to taste transduction was observed as increasing with dose (Appendix A). Interestingly, it has been reported that both olfactory pathways and taste are altered in astronauts, which was previously thought to be due to fluid shifts in the head caused by microgravity conditions [65,66,67]. These data also indicate that space radiation may have some impact on this interesting phenomenon.

#### 2.3.3. GSEA C6: Oncogenic Signatures

To observe specific oncogenic pathways that may change as a function of space radiation exposure, we also included an analysis of the GSEA C6 oncology-specific collection from the molecular signatures database (Figure 7). From the annotation analysis, we discovered that the main oncogenic pathway impacted by space radiation was KRAS (Figure 7A). KRAS is a known oncogene and has been heavily shown in the literature to be involved with lung cancer induced by space radiation [14]. The *k*-means clustering on the annotated gene sets produced the following five clusters: (1) KRAS- and LEF1-related pathways; (2) p53-, mTOR-, KRAS-, and CRX-related pathways; (3) PDGF- and TGFβ-related pathways; (4) TBK1, VEGFA, and ERBB2 pathways; and (5) embryonic stem cells (ESC) pathway (Figure 7B). When observing each cluster as a function of dose, there were interesting trends that can shed light on predicting potential oncogenic risk for longer exposure to space radiation. Thus, the following known oncogenes as well as the most regulated C6 oncogenic signatures were specifically analyzed as a function of dose for all datasets: (1) KRAS, (2) CSR, (3) p53, (4) ESC, (5) TGFβ, (6), VEGFa, and (7) mTOR pathways (Figure 8). These trends as a function of dose across multiple experimental datasets emphasized a need to functionally valid these observations. 

As in the previous sections, we also performed analysis with less stringent statistics with a FDR < 0.05 in only three of the GeneLab Datasets analyzed for C6 gene sets (Appendix A). Because there are fewer gene sets in the C6 database, this analysis started to yield some of the same results as the analysis with the higher thresholds. The GAM fits for each cluster and sub-cluster produced similar results, with one sub-cluster purely associated with TGFβ demonstrating the same trend (Appendix A). Interestingly, one cluster associated with TBK1 did not appear in the higher threshold analysis and demonstrated a clear decrease in as a function of dose (Appendix A). Lastly, a major oncogene MYC [68,69] did not show up in the higher threshold analysis, whereas in this analysis MYC was clearly being activated with increasing dose (Appendix A), indicating that space radiation exposure would increase the potential for carcinogenesis. 

### 2.4. Common Genes as a Function of Dose and Association with Biological Pathways

To look at specific genes that might be common across all datasets, we performed analysis on the log2 fold-change values for all datasets. Nine genes (*COPB2, UBE2D3, LYPLA2, ARF1, COPB1, CAPNS1, PHB2, DAD1, PRPF8*) identified to be common across the spaceflight missions (1.0 to 30 mGy) and ground studies (0.1 to 1.0 Gy) were plotted as a function of dose and average log2 fold-change and fit to a complex polynomial model. These genes, which were expressed across the various tissue types, displayed no significant expression effects from spaceflight up to 100 mGy. At doses higher than 100 mGy, a slight deviation from baseline expression was observed (Figure 9A). The spaceflight studies alone showed slight fluctuations in expression across the identified gene sets (Figure 9B). 

A focused analysis of only the muscle tissue datasets identified a subset of nine different genes that were differentially expressed across the spaceflight studies from 0 to 30 mGy (Figure 9C). These genes displayed dose-response trends. The nine common genes associated with all tissue types were associated with pathways centered on Golgi to endoplasmic reticulum transport and post-translational protein modifications (Figure 9D). In addition, these genes also were predicted to regulated phospholipid activity, immune dysregulation, and embryonic lethality. We observed these pathways to also be involved with our GSEA analysis as a function of dose.

## 3. Discussion

Here, we presented a novel approach in using publicly available transcriptomic data to determine potential health risks and dysregulation of biological pathways as a function of space radiation exposure. The use of NASA’s GeneLab platform [23] allowed us to mine data that might not have otherwise been possible to obtain from individual experimental datasets. The large number of radiation datasets available through this platform allowed the creative combination of different datasets from former experiments to generate new hypotheses [71].

Our hypothesis for this work suspected that there was a tissue independent dysregulation as a function of space radiation-absorbed dose that impacted key biological pathways associated with increased health risks such as induction of cancer. Because the GeneLab Data System contains spaceflight transcriptomic data from mice and tissues flown in low Earth orbit (LEO) for various time durations, we were able to obtain accumulated space radiation doses from 1 to 30 mGy. To obtain doses that will be more relevant to long-term deep space travel [7,14], we combined our analysis with simulated space radiation ground experiments performed in the NASA Space Radiation Laboratory (NSRL) at Brookhaven National Laboratory [24]. These experiments were performed on both mice and in vitro cell cultures with single ions (protons, ^56^Fe, and ^26^Si) at doses from 100 to 1000 mGy. By combining 25 GeneLab datasets over these doses, this work represents the first time a study has provided a comprehensive biological analysis and produced relevant and novel results indicating which relevant biological processes are being regulated as a function of space radiation exposure. These results may be further utilized to generate health risk assessment models and uncover potential targets for mitigation of space radiation effects via novel therapeutics.

We chose to first look at the distribution of mapping errors within the RNA-sequencing results from liver samples by examining whether mice exposure to spaceflight resulted in detectable changes compared to ground controls. We chose to focus on liver tissue datasets from GeneLab with mice exposed to space radiation because on in the GLDS liver is the organ with the largest number of sequenced tissue thus far from rodent research missions [71]. This extremely novel analysis demonstrates how RNA-sequence data can be used to generate useful data other than the standard techniques to generate counts. With a cumulative exposure of 9.20 mGy for GLDS-168 mice, significant increases in A > G and C > T base substitutions were detected, as well as changes in CAC, CGT, GTA, or GTG error rates (Figure 1). However, there were no significant base-specific substitutions detected, and CAC, CGT, GTA, or GTG error rates all decreased significantly in GLDS-242 where mice received a cumulative dose of 8.03 mGy; however, due to the uncertainty inherent in the flight dosimetry, it was not possible to definitively determine whether this was the case. 

A major difference potentially impacting the observed differences between GLDS-168 and GLDS-242 was the animal sacrificing time at sample collection, which occurred during spaceflight for GLDS-168, but was 2 weeks after returning to Earth for GLDS-242. The recovery time for the mice after 2 weeks back on Earth might possibly allow recovery of the base substitutions and error rates observed during exposure to space radiation. The ability to detect single nucleotide variants (SNVs) in RNA and DNA datasets has been performed previously on the basis of the assumption of a homogeneous genomic sample for example from a tumor [70], whereas the sporadic and stochastic nature of HZE exposure is expected to result in random damage throughout the genome [72]. Therefore, approaches to map SNVs on the basis of bulk tissue approaches may not be able to account for the stochastic nature of the SNV damage accumulated in a tissue. Our interpretation of these results was that the spaceflight animals may have experienced more DNA damage/mutation than ground controls, but in the animals that were returned to Earth, the recovery of the animals allowed for repair to take place. 

Next, purely on the basis of statistical analysis, we were able to determine key biological functions that were being dysregulated as a function of space radiation. For example, in the C5 Gene Ontology (GO) terms (Figure 5C), ribosomal pathways were observed as decreasing with dose. Ribosomal assembly consisted of a number of ribosomal protein large (rpl) genes that are necessary in the assembly of the eukaryotic ribosome 60S subunit, as well as the ribosomal protein small genes involved 40S and/or 60S subunit assembly. Indeed, ribosome-related genes have been previously reported to be disrupted following exposure to ionizing radiation [73,74,75]. 

We observed a clear trend of increased activation of mitochondrial-related pathways in multiple different pathway databases, which include both the C5 GO terms (Figure 5C) and C2 curated gene sets (Figure 6C). Mitochondrial gene expression is susceptible to radiation-sensitivity [76]. A number of gene sets in these clusters showed dysregulation of the NADH: ubiquinone oxidoreductase (NDUF) gene family and mitochondria-related function, in agreement with effects of ionizing radiation on the mitochondrial respiratory chain complexes in previous studies [77,78,79]. This is especially relevant to human health, as the NASA Twins Study recently revealed mitochondrial stress and increased mitochondrial debris in the blood [21]. 

A distinct elicitation of inflammatory and immune-related cellular functions was also observed in this analysis with alteration in chemokine, cytokine, and interferon expression (Figure 5C and Figure 6C). Indeed, it has been very well appreciated that ionizing radiation exposure will cause both pro-inflammatory and anti-inflammatory expression as cellular signals attempt to resolve the unique normal tissue radiation injury that occurs both acutely and longer-term [80,81,82]. This immune signaling dysregulation has also been reported to occur when exposed to space radiation [83,84].

We also explored the potential cancer risk as a function of space radiation. The datasets we utilized for this study were from both in vitro and in vivo samples that did not have any cancer present and were not prone to induction of cancer. From our analysis, we observed several interesting results indicative of increased risk for the induction of cancer. In our analysis on the C6 Oncogenic Signature database, we observed that PDGF and TGFβ were clustered together, purely on the basis of the statistics, and both were decreasing with increasing dose (cluster 3 in Figure 7C). This observation is closely tied to previously established findings. In the literature, PDGF and TGFβ are closely related to cancer progression [85]. In addition, it has been shown in the clinic that decreases in PDGF and TGFβ levels after radiotherapy for breast cancer can cause echocardiographic alterations affecting cardiovascular morbidity [86]. We also observed the highest frequency of the KRAS pathway, a known oncogene [87], being impacted as a function dose (Figure 7A). It has been previously described in experiments with single ion species that KRAS can induce lung cancer due to HZE particles [14]. Our tissue-independent analysis did not demonstrate any significant induction of KRAS with increasing dose (cluster 1 in Figure 7C and Figure 8). This might indicate that either previous experiments done with single ions at higher HZE doses did not capture the dose-dependent trend in tissues, or when considering the full spectrum of ions at lower doses KRAS might not be the key driver in inducing tumor growth.

Other oncogenic trends we observed were decreases in CSR, ESC, TGFβ, VEGFA, and mTOR C6 pathway levels as a function of increasing radiation dose (Figure 7 and Figure 8). TGFβ, VEGFa, and mTOR have all previously shown similar trends with increasing dose of simulated space radiation [32,88]. The ESC C6 term is refers to differentiation of embryonic stem cells (ESC) [89]. This rather surprising result indicates that increasing dose of space radiation reduces the efficiency of stem cell differentiation. It has been reported that after ionizing radiation exposure, mitochondria are heavily involved with stem cell development [90]. Exploiting these different pathway databases (i.e., GO, C2, and C6) provided key insights on how these different pathway trends are intertwined.

The CSR C6 term refers to “common serum response” (CSR) specifically related to fibroblast serum [91]. Fibroblast serum response refers to factors released in the serum from fibroblasts involved in assisting with wound healing during tumorigenesis [91]. We can relate the decrease to the CSR pathways to what we observed in the GO analysis (cluster 3 in Figure 5) related to wound healing and VEGF pathways. We also observed with that wound healing should decrease with increasing doses of space radiation. This decrease can also reduce wound healing due to CSR and has shown that it can lead to tumorigenesis [91]. In addition, it has been reported that impairment occurs in space wound healing [92,93], and these data can potentially indicate an increased chance for tumor induction.

These observed trends as a function of space radiation exposure revealed unique pathways that seem to be specifically related to GCR, whereas some pathways from our analysis are known to be universal responses to radiation. The mechanisms of damage are common to highly ionizing radiation regardless of the source [94]; what is unique to chronic space radiation exposure is the combination of highly ionizing particles, low dose, and a low dose rate, which may have implications for DNA repair [17], mitochondrial pathways [16], and lipid pathways [42] as just a few examples of what emerged from our analysis. In addition, we found specific cancer-related pathways that may be unique to space radiation. For example, decreases in TGFβ [32], embryonic stem cells (ESC) [95], and VEGFa seem to be unique to space radiation. Spaceflight studies have previously revealed a plethora of effects that radiation can have at individual doses or over a small range of discreet doses. However, the pooling and meta-analysis of these datasets in this study allowed for a unique visualization and comparison over a larger range of radiation doses. This in turn made it possible to identify trends in physiological response from low to high doses and provided insight into the response to accumulating doses that may occur in long duration space missions or over a few days during solar particle events. 

This technique may also be utilized to determine potential targets for therapeutic intervention. For example, because we report that mitochondrial pathways decrease as a function of space radiation, this argues for additional validation and research on specific mitochondrial targets for countermeasure development. In addition, from this analysis we can extrapolate to specific doses that may pose greater health risks. This analysis can only be done by utilizing a large number of datasets that are available from platforms such as NASA’s GeneLab. The resolution obtained from this analysis can easily be lost once we lower the number datasets that will be used. 

Our analysis has indicated several different health risks that may occur with increasing exposure to space radiation. Through an unbiased technique purely driven by statistics and utilizing a large amount of transcriptomic data from NASA’s GeneLab platform, we were able to identify tissue-independent biological pathways that are dysregulated as a function of space radiation dose. We believe this technique can further be used with other datasets and databases to generate more hypotheses that will assist with determining optimal therapeutic countermeasures and targets that can mitigate space radiation response in humans.

## 4. Materials and Methods 

### 4.1. Chromosomal and Subsitution Error Rate Mapping

Two mouse liver RNA-sequencing datasets (GLDS-168 and GLDS-242) from NASA spaceflight experiments were specifically analyzed for chromosomal or substitution error rates. Compressed fastq files were cleaned using TRIM galore (v0.6.4, https://github.com/FelixKrueger/TrimGalore/blob/master/Docs/Trim_Galore_User_Guide.md) to remove adapters and bases falling below a Q20 Phred quality score. Paired reads were aligned to the mouse mm10 reference genome using bowtie2 (v2.3.4.1,) and the output piped through samtools to generate a bam file. The bam file had read group information added and was coordinate sorted using Picard tools (v2.21.3,). Mapping metrics were calculated using Picard tools and final mismatches (MN flag) extracted with genomic coordinates using Bedtools (v2.26.0, https://bedtools.readthedocs.io/en/latest/). Data were analyzed in R to determine the frequency of mismatches with respect to chromosome number and plotted. Mapping quality data from Picard tools was analyzed using GraphpadPrism (v7.0, GraphPad Software, La Jolla California USA). 

### 4.2. Bioinformatics, Gene Set Enrichment Analysis, and Visulization Using Cytoscape

The transcriptome datasets used for this study were from openly available data housed by NASA’s GeneLab platform (genelab.nasa.gov,). There was a total of 25 GeneLab datasets (GLDS) with 30 experimental comparisons from RNA-sequencing or microarray platforms (Table 1). Data pre-processing was performed as previously published to obtain normalized expression values for the majority of the datasets [32,33,42]. For GLDS-73, human bronchial epithelial cells (HBECs) were exposed to gamma-rays (1 Gy or 3 Gy), iron (^56^Fe), or silicon (^26^Si) particles (0.5 Gy or 1 Gy each). RNA extraction was executed 1 hr after exposure and the transcriptome data were evaluate by microarray technology using the Illumina HumanWG-6 V2 BeadChip (Illumina, Inc.) [52]. The raw data files were normalized using the quantile method and then background subtraction was performed using R Package PIMENTo (v1.0). 

Fold-change values between spaceflight/radiation exposure versus ground/control samples were calculated and used for gene set enrichment analysis (GSEA,) with three molecular signature dataset collections (C2, C5, and C6 gene sets) [54]. These results were imported into Cytoscape (v3.7) for further visualization and interpretation by the Enrichment Map tool [96]. Only gene set nodes with a false-detection rate (FDR) < 0.25 in at least 50% of the total datasets analyzed were considered for network evaluation. To reduce the complexity of the newly developed networks, we utilized the AutoAnnotate tool (v1.3.2, http://baderlab.org/Software/AutoAnnotate) in Cytoscape to group-related gene set nodes together. The newly annotated gene sets were then used for further analysis.

### 4.3. Principal Component Analysis, t-Distributed Stochastic Neighbor Embedding, k-Means Clustering, Heatmaps, and Generalized Additive Model Fits

The GSEA-produced nominal enrichment scores (NES) were compared using principle component analysis (PCA) plots from the R-program pca3d (v0.10, https://cran.r-project.org/web/packages/pca3d/pca3d.pdf) [97], and t-distributed stochastic neighbor embedding (t-SNE) plots using R-program Rtsne (v0.15, https://cran.r-project.org/web/packages/Rtsne/Rtsne.pdf). Further characterization was performed by hierarchical *k*-means clustering on the NES. Heatmaps were generated using the R-program pheatmap (v1.0.12, https://cran.r-project.org/web/packages/pheatmap/pheatmap.pdf). The auto-annotated gene sets for each *k*-means cluster were then plotted with the NES versus the dose for associated for each GeneLab dataset (Table 1) with R-program ggplot2 (v3.2.1,). Data points were then fit using a generalized additive model (GAM) [55,56] across all dose ranges from 1 to 1000 mGy with R-program mgcv (v1.8-28, https://cran.r-project.org/web/packages/mgcv/mgcv.pdf)

### 4.4. Analysis on Common Individual Genes with Fold-Change Values and Fits as a Function of Dose

A total of 19 spaceflight studies and 3 ground studies were selected from the GeneLab database. Table 1 summarizes the datasets including the tissue, species, the radiation type, and the measured dose. From each study, the fold-change value between irradiated and control for each gene was calculated. For spaceflight studies, the log_2_ fold-change was computed as the ratio of the spaceflight measurement to a ground control value. In studies where terrestrial measurements for ground control values were made in both an animal enclosure module (AEM, also referred to as the Rodent Habitat) and a vivarium, the AEM ground control value was used due to the similar module conditions experienced by tissue in the spaceflight AEMs. For the terrestrial studies that simulated space radiation, the experimental values were the samples that were irradiated with a specific ion (i.e., ^56^Fe, ^26^Si, or proton) compared to the sham unirradiated samples closest to the time of irradiation. Across all of the studies, the most common genes probed were chosen, which produced nine genes present with all the GLDS sets used for this manuscript. A polynomial of degree 2 was fit to the data in the range 1–1000 mGy for all datasets. For fits involving only spaceflight samples and only muscle tissues, due to the smaller range of doses up to 30 mGy, data were fit using a polynomial of degree 1. All fits were performed using a chi-square method via CERN’s ROOT toolkit (v.6.18/02,) [98]. The nine genes were entered in ToppCluster [99] to determine the predicted pathways and functions these genes will impact. The results from ToppCluster were imported into Cytoscape, where the predicted functions were displayed as a network.

## 5. Conclusions

This work is an example of how GeneLab’s rich database of omics experiments with animal models can be exploited to better understand the effects of ionizing radiation and the resulting potential risks to humans. This large-scale analysis of multiple transcriptome datasets including RNA-sequencing and microarray datasets provides a valuable examination of dose-dependent perturbations of cellular function across multiple tissue types in animal models. Individually, these experiments were limited by flight duration, resulting in a single accumulated dose from a micture of particles or, in ground-based studies, exposure to a single particle species. When analyzing multiple molecular databases using our approach to many omics datasets, a more comprehensive analysis of significantly impacted biological factors can be obtained than was previously possible.

## Figures and Tables

**Figure 1 cancers-12-00381-f001:**
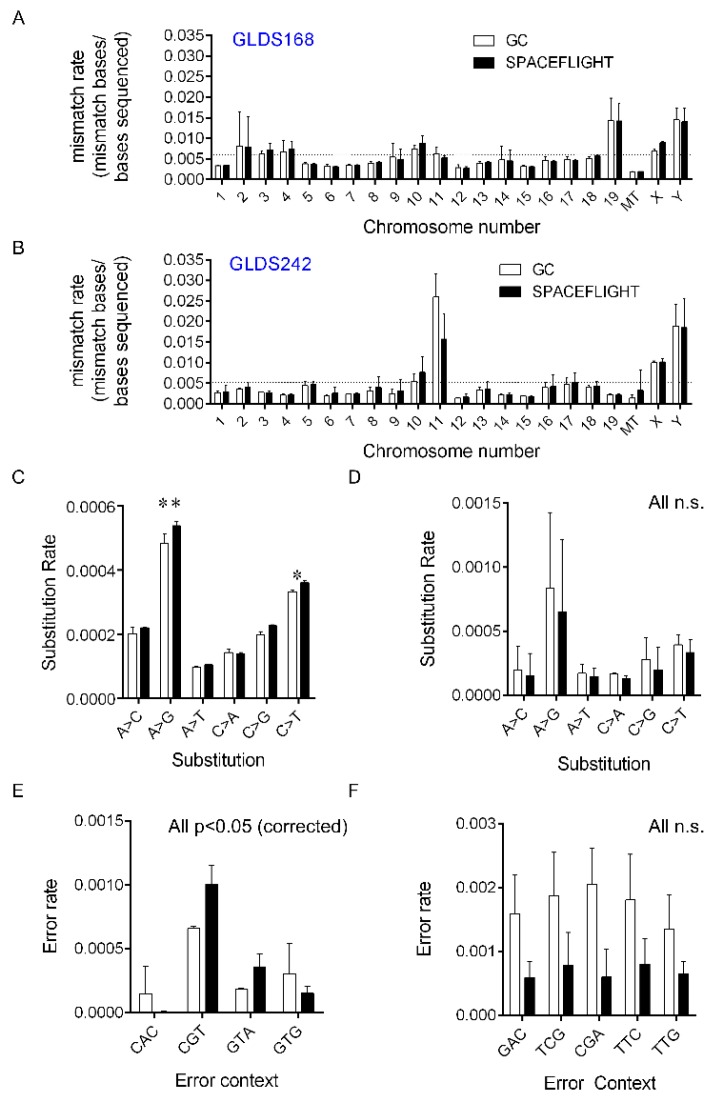
RNA-sequencing error rates in mouse livers from GLDS-168 (left column) and GLDS-242 (right column) experiments for spaceflight (black bars) compared to ground control samples (white bars). Error rates were examined as follows: (**A**,**B**) mismatch rates across chromosomes normalized by the number of bases sequenced; (**C**,**D**) single base substitution rates; (**E**,**F**) CAC, CGT, GTA, or GTG error rates. Significance indicated by * *p*-value < 0.05 and ** *p*-value < 0.01.

**Figure 2 cancers-12-00381-f002:**
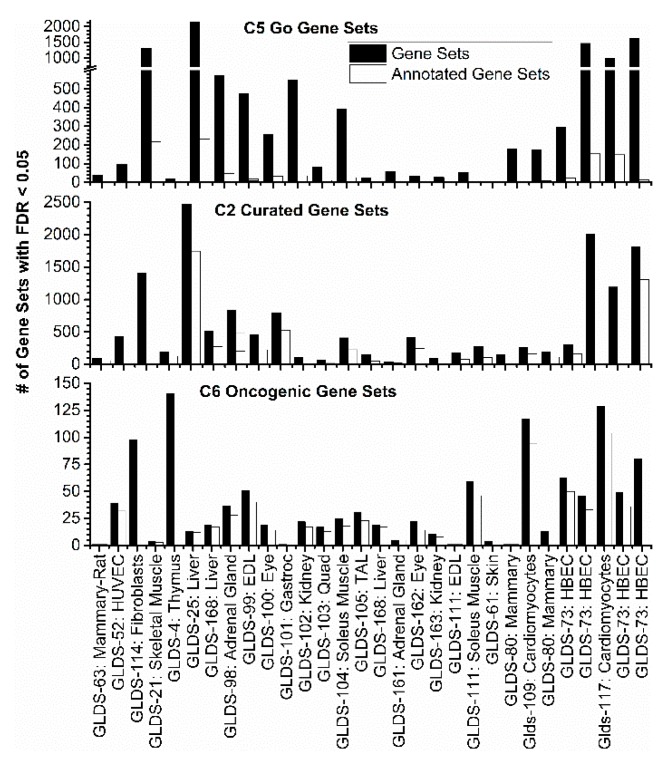
The number of statistically significant gene sets from gene set enrichment analysis (GSEA) with a false discovery rate (FDR) < 0.05 for each dataset for the major molecular signatures database collections (i.e., C5, C2, and C6). The black bars represent the unaltered genes and the white bars represent the annotated gene sets.

**Figure 3 cancers-12-00381-f003:**
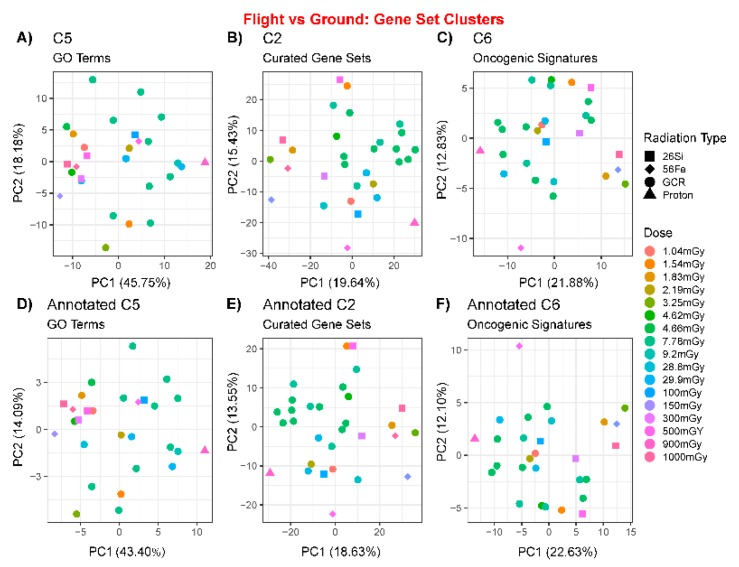
Principal component analysis (PCA) plots on GLDS datasets using the gene set enrichment analysis (GSEA) nominal enrichment scores before (top row) and after (bottom row) auto-annotation using Cytoscape’s enrichment map software. GSEA gene sets used were as follows: (**A**,**B**) C5 Gene Ontology (GO) terms, (**C**,**D**) C2 curated terms, and (**E**,**F**) C6 oncogenic signature terms. Each data point is represented by experimental radiation dose (color) or radiation type (shape).

**Figure 4 cancers-12-00381-f004:**
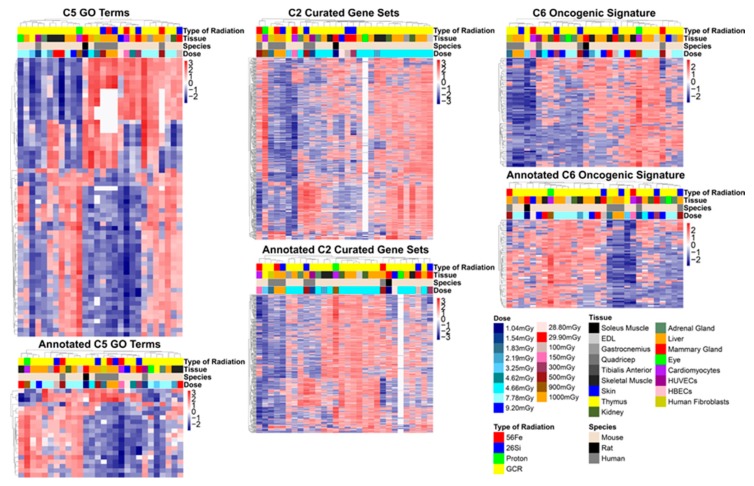
Heatmaps on all gene sets and also annotated gene sets for each molecular signature database collection. Euclidean clustering was used for clustering the rows and columns.

**Figure 5 cancers-12-00381-f005:**
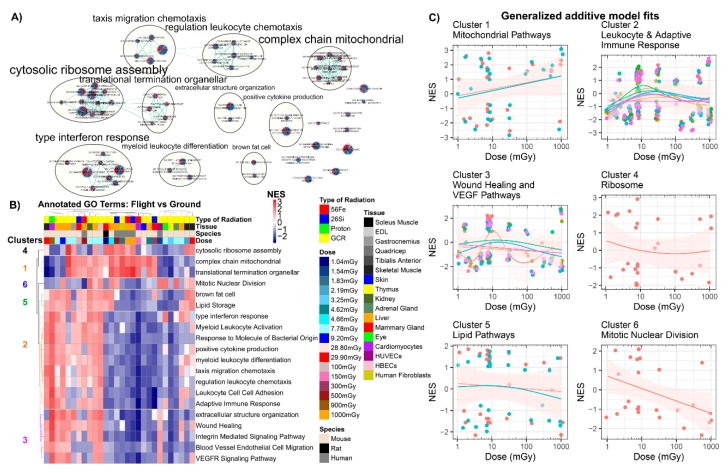
Analysis of all GeneLab datasets utilized for direct comparisons with Gene Ontology (GO) GSEA analysis. (**A**) Auto-annotated GO GSEA terms from Cytoscape’s enrichment mapper. Each node represents one specific GO term with each wedge representing one GeneLab datasets utilized for this manuscript. The yellow circles represent the auto-annotated go terms with common related pathways. (**B**) Heatmap with *k*-means clustering for the specific GO pathways. Six specific pathways were found to be clustered together through *k*-means clustering. (**C**) Scatter plots comparing the normalized enrichment scores (NES) to dose in milligrays and fitting with a generalized additive model (GAM) on each cluster (represented in (**C**)). Each panel represents one of the clusters and the color-coded lines represent GAM fits performed on each GO-annotated term in the cluster. The circles in the plots represent the gene sets and the pink shade around the GAM fits represent the standard error.

**Figure 6 cancers-12-00381-f006:**
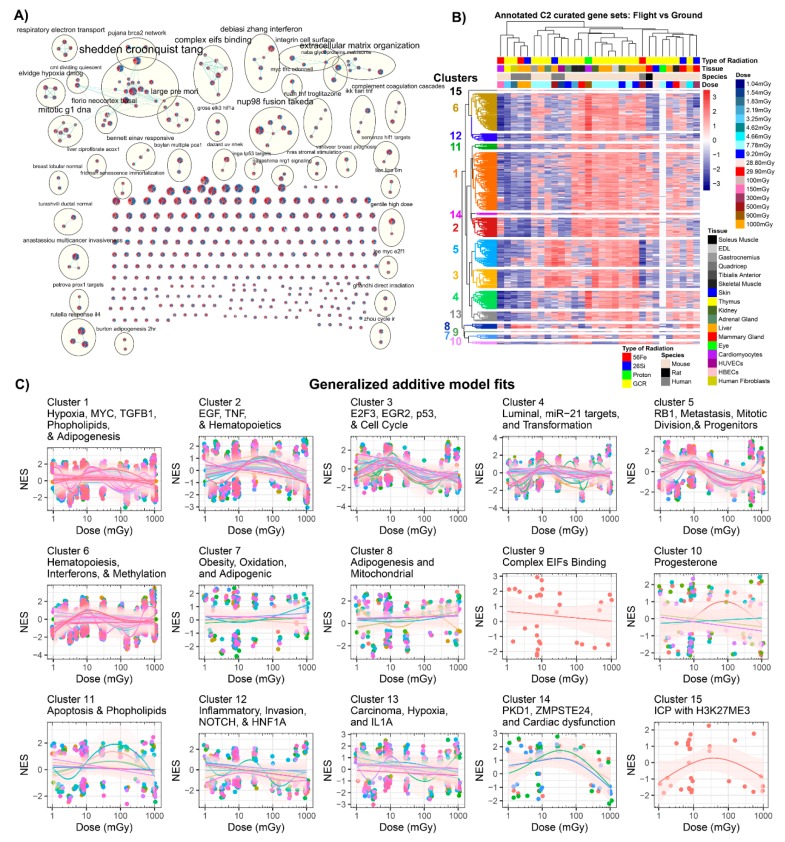
Analysis of all GeneLab datasets utilized for direct comparisons with the GSEA C2 curated collection analysis. (**A**) Auto-annotated C2 GSEA terms from Cytoscape’s enrichment mapper. Each node represents one specific C2 term with each wedge representing one GeneLab dataset utilized for this manuscript. The yellow circles represent the auto-annotated GO terms with common related pathways. (**B**) Heatmap with *k*-means clustering for the specific C2 pathways. A total of 15 specific pathways were found to be clustered together through *k*-means clustering. (**C**) Scatter plots comparing the normalized enrichment scores (NES) to dose in mGy and fits with a generalized additive model (GAM) on each cluster (represented in (**B**)). Each panel represents one of the clusters and the color-coded lines represent GAM fits performed on each C2-annotated term in the cluster. The circles in the plots represent the gene sets and the pink shade around the GAM fits represent the standard error.

**Figure 7 cancers-12-00381-f007:**
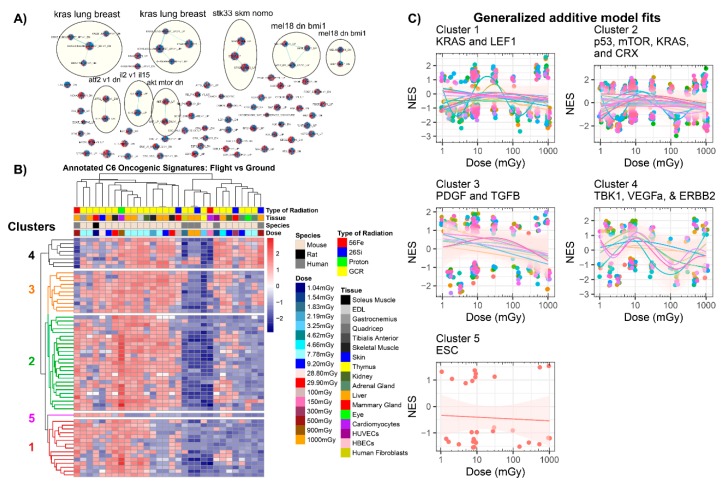
Analysis of all GeneLab datasets utilized for direct comparisons with the GSEA C6 oncology signature collection analysis. (**A**) Auto-annotated C6 GSEA terms from Cytoscape’s enrichment mapper. Each node represents one specific C6 term with each wedge representing one GeneLab dataset utilized for this manuscript. The yellow circles represent the auto-annotated GO terms with common related pathways. (**B**) Heatmap with *k*-means clustering for the specific C6 pathways. Five specific pathways were found to be clustered together through *k*-means clustering. (**C**) Scatter plots comparing the normalized enrichment scores (NES) to dose in milligrays and fits with a generalized additive model (GAM) on each cluster (represented in (B)). Each panel represents one of the clusters and the color-coded lines represent GAM fits performed on each C6-annotated term in the cluster. The circles in the plots represent the gene sets and the pink shade around the GAM fits represent the standard error.

**Figure 8 cancers-12-00381-f008:**
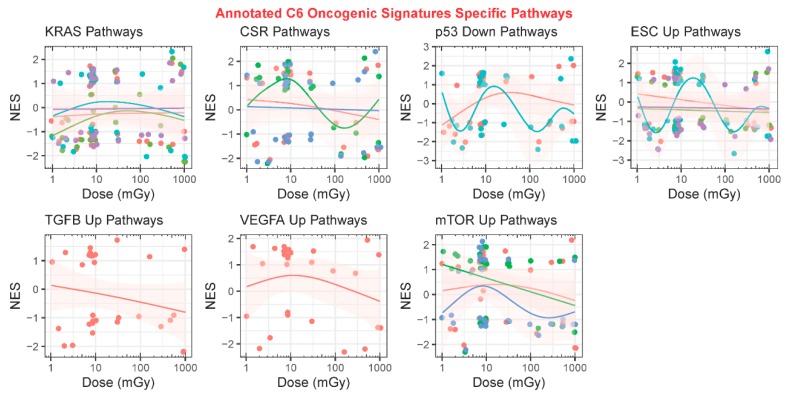
Analysis of all GeneLab datasets utilized for direct comparisons with the GSEA C6 oncology signature collection analysis. Scatter plots comparing the normalized enrichment scores (NES) to dose in milligrays and fits with a generalized additive model (GAM) on specific C6 oncogenes. Each panel represents one specific oncogene and the color-coded lines represent GAM fits performed on the multiple C6 pathways associated with that oncogene. The circles in the plots represent the gene sets and the pink shade around the GAM fits represent the standard error.

**Figure 9 cancers-12-00381-f009:**
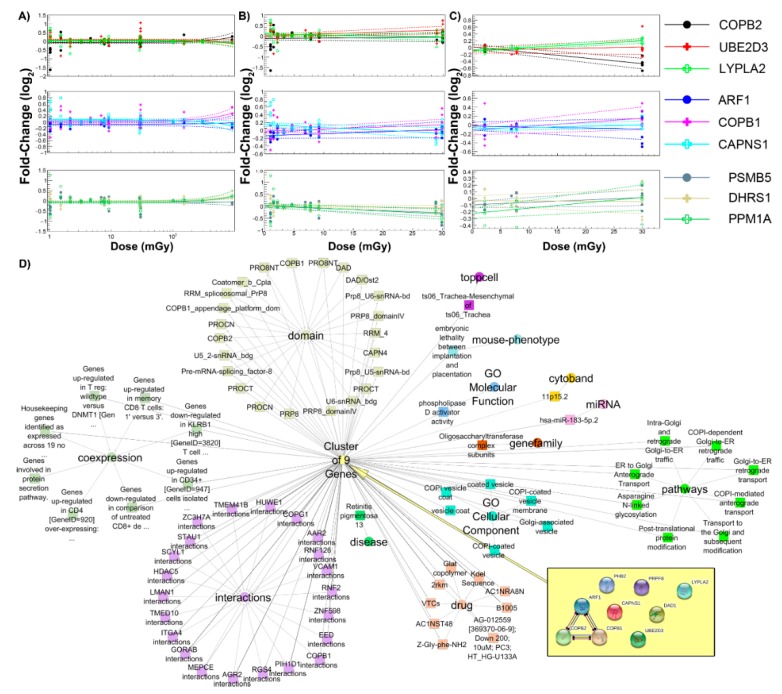
Common genes across all datasets. (**A**) Log2 fold-change with respect to dose for a subset of 9 genes that appeared across all 19 spaceflight studies and 3 ground studies. A polynomial of degree 2 was fit to each dose response in the range 1–1000 mGy. (**B**) Log2 fold-change with respect to dose for a subset of nine genes that appeared across spaceflight studies in the range 1–30 mGy. A polynomial of degree 1 was fit to each dose response. (**C**) Log2 fold-change with respect to dose for a subset of nine genes that appeared across all eight spaceflight studies specifically related to the muscle tissues. A polynomial of degree 1 was fit to each dose response in the range 1–30 mGy. The dashed lines denote the +/-1 sigma errors on the fitted polynomials. (**D**) The functional impact of the nine common genes determined by ToppCluster [70] and displayed using Cytoscape.

**Table 1 cancers-12-00381-t001:** Summary of the NASA GeneLab datasets for samples exposed to cosmic radiation during spaceflight or analogous ground-based radiation experiments curated for analysis.

GLDS	Condition	Dose (mGy)	Radiation Type *	Species	Tissue/Cell Culture	Ref.
63	Spaceflight	1.04	LEO	Rat	Mammary	[25,26,27]
52	Spaceflight	1.54	LEO	Human	Endothelial cells (HUVEC)	[28,29]
114	Spaceflight	1.83	LEO	Human	Fibroblast cells (AG01522)	[30]
21	Spaceflight	2.19	LEO	Mouse (C57BL/6)	Skeletal muscle	[31,32,33]
4	Spaceflight	3.25	LEO	Mouse (C57BL/6)	Thymus	[32,34]
47	Spaceflight	4.62	LEO	Mouse (C57BL/6)	Liver	[35]
25	Spaceflight	4.66	LEO	Mouse (C57BL/6)	Liver	[32,33,36,37,38,39,40,41,42]
98	Spaceflight	7.78	LEO	Mouse (C57BL/6)	Adrenal gland	[32]
99	Spaceflight	7.78	LEO	Mouse (C57BL/6)	Muscle (extensor digitorum longus)	[32]
100	Spaceflight	7.78	LEO	Mouse (C57BL/6)	Eye	[43]
101	Spaceflight	7.78	LEO	Mouse (C57BL/6)	Muscle (gastrocnemius)	[32]
102	Spaceflight	7.78	LEO	Mouse (C57BL/6)	Kidney	[32]
103	Spaceflight	7.78	LEO	Mouse (C57BL/6)	Muscle (quadriceps)	[32]
104	Spaceflight	7.78	LEO	Mouse (C57BL/6)	Muscle (soleus)	[32]
105	Spaceflight	7.78	LEO	Mouse (C57BL/6)	Muscle (tibialis anterior)	[32]
168	Spaceflight	7.78	LEO	Mouse (C57BL/6)	Liver	[42,44]
242	Spaceflight	8.03	LEO	Mouse (C57BL/6)	Liver	[45]
161	Spaceflight	9.20	LEO	Mouse (BALB/C)	Adrenal gland	[46]
162	Spaceflight	9.20	LEO	Mouse (BALB/C)	Eye	[47]
163	Spaceflight	9.20	LEO	Mouse (BALB/C)	Kidney	[48]
168	Spaceflight	9.20	LEO	Mouse (BALB/C)	Liver	[42,44]
61	Spaceflight	28.80	LEO	Mouse (C57BL/6)	Skin	[32,49]
111	Spaceflight	29.90	LEO	Mouse (C57BL/6)	Muscle (extensor digitorum longus)	[5,32,33]
111	Spaceflight	29.90	LEO	Mouse (C57BL/6)	Muscle (soleus)	[5,32,33]
80	Ground	100	Silicon	Mouse (BALB/C)	Mammary tissue	[50]
109	Ground	150	Iron	Mouse (C57BL/6)	Cardiomyocytes	[29,51]
80	Ground	300	Silicon	Mouse (BALB/C)	Mammary tissue	[50]
73	Ground	500	Iron	Human	Bronchial epithelial cells	[52]
73	Ground	500	Silicon	Human	Bronchial epithelial cells	[52]
117	Ground	900	Proton	Mouse (C57BL/6)	Cardiomyocytes	[29,51]
73	Ground	1000	Iron	Human	Bronchial epithelial cells	[52]
73	Ground	1000	Silicon	Human	Bronchial epithelial cells	[52]

* LEO: low Earth orbit radiation for flight experiments is due to a combination of cosmic radiation (protons, galactic cosmic rays (GCR), and heavier ions) as well as protons trapped in Earth radiation belts.

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
