# Peer review of "NASA GeneLab Platform Utilized for Biological Response to Space Radiation in Animal Models"

_cancers, 2020, doi:10.3390/cancers12020381_

Round 1

Reviewer 1 Report

McDonald et al present an interesting discussion of an -omics approach to understanding the effect of cosmic radiation on biological processes. Using both spaceflight and ground-based radiation experimental samples, they present a comprehensive look at transcriptomic signatures using GSEA analysis. 

Their statistical analysis is well described and the discussion is comprehensive, but support for the overall novelty and significance of these findings is somewhat lacking. GSEA analysis presents high level predictions of implicated gene sets, but does not support functional roles for given genes. The majority of the canonical pathways and GO signatures identified involve fundamental biological processes, many of which would have been predicted to change in response to radiation exposure. 

Are these signatures similar to traditional radiation exposure, or is this unique to GCR?

With so many fundamental systems implicated, how feasible is predicting or designing therapeutic intervention?

What are the limitations of this approach? 

Minor Edits:

Page 2 Line 87 - check plural agreement (datatsets...presents). Should be datasets...present.

Page 2 Line 89-90 - shift comma. Should read "...datasets in GeneLab, relevant to spaceflight, was undertaken to uncover..."

Page 4 Line 125-136 - check plural agreement (mutations...unless it is). Should be mutations...unless they are.

Page 14 Line 387 - "...as a function of increasing [radiation] dose."

Page 14 Line 395-399. The phrasing of this discussion point is somewhat confusing. I would recommend separating the idea of wound healing and tumorigenesis into conceptual sections within the paragraph. OR provide a more robust commentary on the link between wound healing and tumorigenesis.

Figures: All are difficult to read at the current size - consider increasing font size in original figure files.

Author Response

Dear Reviewer,

We have provided our response to your comments in red font below each of your comments. We thank you for your insightful comments and believe that the edits we have made based on your comments have made this a stronger manuscript.

Sincerely,

Afshin Beheshti

McDonald et al present an interesting discussion of an -omics approach to understanding the effect of cosmic radiation on biological processes. Using both spaceflight and ground-based radiation experimental samples, they present a comprehensive look at transcriptomic signatures using GSEA analysis. 

Their statistical analysis is well described and the discussion is comprehensive, but support for the overall novelty and significance of these findings is somewhat lacking. GSEA analysis presents high level predictions of implicated gene sets, but does not support functional roles for given genes. The majority of the canonical pathways and GO signatures identified involve fundamental biological processes, many of which would have been predicted to change in response to radiation exposure. 

We thank the reviewer for their insightful comments. This exact fact that this analysis provides pathways that are commonly related to radiation exposure, shows a type of positive control. Also the fact that this analysis provides common pathways being impacted as a function of dose independent of ion or GCR and tissue type, is an intriguing insight on implicating these pathways to be dysregulated purely due to space radiation exposure. It has been shown that ion type sometime impacts the biology differently. We believe that the novelty of this tissue independent and ion independent pathways has not previously been shown at such a large scale of data with a comprehensive examination in this lower dose range (1mGy to 30mGy).

Are these signatures similar to traditional radiation exposure, or is this unique to GCR?

Some signatures are similar to traditional radiation exposure such as decrease mitotic nuclear division with increasing dose. Other pathways such as increased mitochondrial functions as a function of dose seems to be unique to GCR. We have provided addition discussion to address this comment by the reviewer at the end of the discussion section. [lines 440-453]

With so many fundamental systems implicated, how feasible is predicting or designing therapeutic intervention?

This technique can be utilized to determine novel pathways to target based on the most pronounced trends. For example the mitochondrial pathways are decreasing in function as a function of dose. A researcher can then use this knowledge to further validate and explore mitochondrial based therapy. We have also added a brief discussion regarding this comment at the end of the discussion section [lines 454-460].

What are the limitations of this approach? 

If others perform this analysis with limited datasets the resolution for all the specific trends might not appear. For this manuscript we utilized NASA’s GeneLab platform which houses many omics datasets that allows us to perform such analysis. We have added a brief discussion at the end of the discussion section discussing this limitation.

Minor Edits:

Page 2 Line 87 - check plural agreement (datatsets...presents). Should be datasets...present.

We have made this change.

Page 2 Line 89-90 - shift comma. Should read "...datasets in GeneLab, relevant to spaceflight, was undertaken to uncover..."

We have made this change.

Page 4 Line 125-136 - check plural agreement (mutations...unless it is). Should be mutations...unless they are.

We have made this change.

Page 14 Line 387 - "...as a function of increasing [radiation] dose."

We have made this change.

Page 14 Line 395-399. The phrasing of this discussion point is somewhat confusing. I would recommend separating the idea of wound healing and tumorigenesis into conceptual sections within the paragraph. OR provide a more robust commentary on the link between wound healing and tumorigenesis.

We have separated the paragraph as the reviewer as suggested and also rewrote this paragraph to become more clear.

Figures: All are difficult to read at the current size - consider increasing font size in original figure files.

We have modified the figures as the reviewer has suggested.

Reviewer 2 Report

The goal of this study was to use publicly available datasets to determine potential modification / dysregulation of biological pathways as a function of space radiation exposure. The authors tried to examine the biological responses of different models (human, murine cell cultures and animal tissues) as a function of dose of cosmic radiations (1 mGy to 30 mGy) or simulated space radiation (protons, 56Fe, and 26Si at doses from at 100 mGy to 1000 mGy). A meta-analysis of transcriptomic datasets in GeneLab, relevant to spaceflight was performed.

According to these omics datasets, the authors compare different statistical parameters and biological responses, using global and large analysis tools. Most of the presented results give response-signatures related to radiation quality and doses. Molecular pathways, supposed to be impacted by radiations are proposed. The authors finally discuss some of the impacted biological pathways associated with space radiations, and compare these transcriptomic data with known response of cells to radiations.

This huge work is well introduced and well described. Such study could give interesting results / ideas indicating which biological processes are being regulated as a function of space radiation exposure.

As a suggestion, I would recommend to the authors to try to present the results with larger plots, allowing the reader to see the results. Here the figures are so small that it is almost impossible to see anything.

Author Response

Dear Reviewer,

We have provided our response to your comments in red font below each of your comments. We thank you for your insightful comments and believe that the edits we have made based on your comments have made this a stronger manuscript.

Sincerely,

Afshin Beheshti

The goal of this study was to use publicly available datasets to determine potential modification / dysregulation of biological pathways as a function of space radiation exposure. The authors tried to examine the biological responses of different models (human, murine cell cultures and animal tissues) as a function of dose of cosmic radiations (1 mGy to 30 mGy) or simulated space radiation (protons, 56Fe, and 26Si at doses from at 100 mGy to 1000 mGy). A meta-analysis of transcriptomic datasets in GeneLab, relevant to spaceflight was performed.

According to these omics datasets, the authors compare different statistical parameters and biological responses, using global and large analysis tools. Most of the presented results give response-signatures related to radiation quality and doses. Molecular pathways, supposed to be impacted by radiations are proposed. The authors finally discuss some of the impacted biological pathways associated with space radiations, and compare these transcriptomic data with known response of cells to radiations.

This huge work is well introduced and well described. Such study could give interesting results / ideas indicating which biological processes are being regulated as a function of space radiation exposure.

As a suggestion, I would recommend to the authors to try to present the results with larger plots, allowing the reader to see the results. Here the figures are so small that it is almost impossible to see anything.

We thank the reviewer for this insightful comments and we have modified the figures as the reviewer has suggested.